# Targeting the Ubiquitin–Proteasome System and Recent Advances in Cancer Therapy

**DOI:** 10.3390/cells13010029

**Published:** 2023-12-22

**Authors:** Daniela Spano, Giuliana Catara

**Affiliations:** 1Institute for Endocrinology and Experimental Oncology “G. Salvatore”, National Research Council, Via Pietro Castellino 111, 80131 Naples, Italy; 2Institute of Biochemistry and Cell Biology, National Research Council, Via Pietro Castellino 111, 80131 Naples, Italy

**Keywords:** ubiquitination, proteasome-mediated degradation, cancer, therapeutic approaches, synthetic lethality, protein post-translational modifications

## Abstract

Ubiquitination is a reversible post-translational modification based on the chemical addition of ubiquitin to proteins with regulatory effects on various signaling pathways. Ubiquitination can alter the molecular functions of tagged substrates with respect to protein turnover, biological activity, subcellular localization or protein–protein interaction. As a result, a wide variety of cellular processes are under ubiquitination-mediated control, contributing to the maintenance of cellular homeostasis. It follows that the dysregulation of ubiquitination reactions plays a relevant role in the pathogenic states of human diseases such as neurodegenerative diseases, immune-related pathologies and cancer. In recent decades, the enzymes of the ubiquitin–proteasome system (UPS), including E3 ubiquitin ligases and deubiquitinases (DUBs), have attracted attention as novel druggable targets for the development of new anticancer therapeutic approaches. This perspective article summarizes the peculiarities shared by the enzymes involved in the ubiquitination reaction which, when deregulated, can lead to tumorigenesis. Accordingly, an overview of the main pharmacological interventions based on targeting the UPS that are in clinical use or still in clinical trials is provided, also highlighting the limitations of the therapeutic efficacy of these approaches. Therefore, various attempts to circumvent drug resistance and side effects as well as UPS-related emerging technologies in anticancer therapeutics are discussed.

## 1. Introduction

Proteins play a key role in maintaining cellular homeostasis; therefore, any perturbation affecting protein abundance, localization, enzymatic activity or protein–protein interaction can lead to the dysfunction of cellular processes and the onset of various pathologies. To coordinate such diverse physiological functions, cells activate different programs that involve the tight tuning of gene expression followed by protein post-translational modifications (PTMs) [1]. Among these, ubiquitination is one of the most widespread PTMs in the cell. It is a reversible reaction involving the covalent addition of the 76-amino-acid ubiquitin (Ub) protein to substrates via a sequential enzymatic reaction [2,3] catalyzed by E1 Ub-activating, E2 Ub-conjugating and E3 Ub-ligating enzymes [4].

The reaction can be further classified as monoubiquitination or polyubiquitination, based on the transfer of a single or multiple ubiquitin moiety(s) in response to different stimuli. Ubiquitinated targets may undergo two different fates, namely proteasomal degradation or signaling pathway modulation. In the latter case, once the cellular response has been achieved, specific proteases called deubiquitinases (DUBs), which cleave ubiquitin from substrates or ubiquitin chains, reverse the reaction, thereby quenching ubiquitination signaling [5,6].

Ubiquitination is highly conserved among eukaryotes, where it modulates a high number of cellular functions by coordinating a large part of the signaling networks [3,7,8,9,10,11,12]. The discovery of ubiquitin protein-based modifications in the 1980s linked ATP-dependent ubiquitination of substrates to their degradation by the 26S proteasome [13]. Indeed, the ubiquitin–proteasome system (UPS) regulates the degradation of misfolded and aggregated proteins as well as protein turnover [14]. It is now widely accepted that ubiquitination also plays a key role in the control of several fundamental biological processes, including the cell cycle [15], apoptosis [16], autophagy [11,14], epigenetics [17], as well as NF-kB and T-cell receptor signaling [18,19,20,21], and DNA repair and transcription [22,23,24], to name a few. Similarly, deubiquitinases modulate diverse cellular events, including the cell cycle [25], apoptosis [26], receptor signaling [16,21], gene transcription [24] and DNA repair pathways [23]. The ubiquitination system is also conserved throughout evolution and modulates a wide range of biological functions in bacteria and viruses, where it is emerging as a major player in the bacterial pathogenic mechanisms of infectious diseases [27,28,29,30,31,32]. Ubiquitination-related enzymes have been found in *Yersinia pseudotuberculosis* [33], *Salmonella typhimurium* [34,35], *Escherichia coli* [36], *Legionella pneumophila* [37,38,39,40], *Streptococcus pyogenes* [41], *Shigella flexneri* [42], *Chlamydia trachomatis* [43], where they support the host–pathogen molecular interaction. As a result, bacterial effectors interfere with the host’s ubiquitination machinery, leading to the impairment of the host’s ubiquitin-dependent processes [31,44]. Therefore, a deeper understanding of the ubiquitin-mediated host–pathogen interplay may contribute to the design and implementation of alternative therapeutic approaches for unmet needs in infectious diseases.

Ubiquitination also plays a role in virus–host interactions, where the activity of specific viral proteases, exemplified by the papain-like proteases (PLpros), allows viruses to evade the host’s immune response, as demonstrated by Middle East respiratory syndrome virus (MERS), severe acute respiratory syndrome virus (SARS-CoV) and SARS-CoV-2 virus [45]. PLpro cleaves ubiquitin, interferon-stimulated gene 15 (ISG15) or neural precursor cell-expressed developmentally downregulated protein 8 (NEDD8) from substrates thus impairing the host’s functions [45]. Recent studies have shown that the three PLpros have different substrate specificities that can be exploited for the design of selective inhibitors to limit the diverse viral infections [46,47].

Therefore, the balance between ubiquitination and deubiquitination is important for the maintenance of cellular homeostasis. In humans, genetic alterations, abnormal expression or dysfunction of E3 ubiquitin ligases and DUBs determine the onset and progression of many pathological conditions such as neurodegenerative diseases, immune-related pathologies and cancer [8,48,49,50]. For these reasons, the study of the ubiquitination machinery, as well as the molecular mechanisms underlying its activity and regulation, has been an area of intense research over the last two decades with the aim of developing new therapeutic approaches to treat such pathological conditions [50,51,52,53].

The role of ubiquitination in cancer has gained attention in the last two decades due to the efficacy of proteasome inhibition in the clinic. Bortezomib-based anticancer therapies were first approved by the Food and Drug Administration (FDA) in 2003 for the treatment of relapsed multiple myeloma [54,55] and then were extended to hematological malignancies [56,57]. The reversible inhibition of the 26S proteasome leads to the accumulation of regulatory proteins with the consequent entry of cells into apoptosis. However, limitations due to drug resistance and side effects have increased the need to discover new targets to be drugged, leading to the consideration of ubiquitination machinery enzymes as new opportunities for the development of antitumor approaches.

This perspective article summarizes the key findings on the ubiquitination reaction and reviews the role of ubiquitination in cancer. It discusses the rationale behind cancer therapeutic strategies based on targeting the ubiquitination machinery, highlighting the FDA-approved pharmacological interventions that have entered the clinic, as well as those that are still in clinical trials and whose therapeutic efficacy has been evaluated. Finally, several issues that need to be addressed to achieve effective anticancer therapeutics for clinical management are considered as future perspectives.

## 2. The Ubiquitination Reaction

### 2.1. Enzymes Involved in Ubiquitination Modification

The ubiquitination reaction is catalyzed by the sequential reactions of three enzymes: (i) the E1 Ub-activating enzyme (E1), which activates ubiquitin in an adenosine triphosphate-dependent manner and transfers it to the E2 Ub-conjugating enzyme; (ii) the E2 Ub-conjugating enzyme (E2), whose catalytic cysteine forms a thioester bond with the C-terminal carboxyl group of Ub; (iii) the E3 Ub protein ligase (E3), which mediates the final step of ubiquitin transfer with the formation of an isopeptide bond between the lysine ε-amino group of the substrate and the C-terminal carboxyl group of Ub in the canonical ubiquitination code [2], as discussed in Section 3. Alternatively, the E2s directly transfer the ubiquitin molecule to the substrate after formation of an E2-E3-substrate multicomplex [58,59] (Figure 1).

The bioinformatic analysis of the human genome has shown that there are 2 E1 enzymes, 40 E2 enzymes and about 1000 E3 ligases [2]. These can combine to form different E2-E3 heterocomplexes, each playing an important role in the length and topology of the ubiquitin chain. For example, the E2 enzyme UbcH5c catalyzes a Lys-48-linked chain when in complex with E6-AP, whereas it produces a Lys-6-linked chain when interacting with BRCA1/BARD1. For the sake of simplicity, we will refer to E3 ligases because they are the molecular targets of most of the therapeutic strategies discussed in Section 5 and Section 6.

E3s recognize, interact with and ubiquitinate substrates in a temporally and spatially regulated manner, leading to the assembly of diverse polyubiquitin chain structures and greatly expanding ubiquitination signaling.

Based on the structural homology and mechanism of action, E3s are generally classified into four types: (1) HECT (homologous to the E6AP carboxyl terminus) type, (2) RING (really interesting new gene) type, (3) U-box type and (4) RBR (RING-in-between-RING) type [60,61], which show low sequence similarity and large differences in composition [60,62].

The HECT E3 ubiquitin ligase family represents the majority of E3 enzymes that have been characterized to date. HECT-type E3s accept activated ubiquitin from an E2, forming a thioester intermediate, before transferring ubiquitin to the substrates [63]. Based on the structural organization of the N-terminal domain, HECT-type enzymes are further classified into three subfamilies: (i) Nedd4, which comprises 9 members; (ii) HERC, which comprises 6 members; and (iii) other HECTs, which comprise 13 members.

RING E3 ligases are the largest of the four classes, with more than 600 different members expressed in human cells [58]. They are characterized by the presence of the RING domain, which allows them to interact both with the target and E2 conjugating enzymes, thus catalyzing the direct attachment of ubiquitin to the target. RING E3 ligases are divided into two major subgroups: (i) monomeric RING fingers and (ii) multi-subunit E3 ligases. Monomeric RING E3 ligases can modify cellular targets as well as themselves, such as COP1, MDM2 and TRAF6. Multi-subunit E3 ligases, such as the Cullin RING ligases (CRLs), are a heterogeneous class of ubiquitin ligases characterized by several common structural features that are generally regulated by different modifications such as autoubiquitination, neddylation, phosphorylation and interaction with small molecules [64].

U-box E3 ligases, which contain a conserved C-terminal U-box domain of approximately 70 amino acid residues from yeast to humans, are the smallest group of enzymes involved in the control of proteasomal degradation efficiency [59]. Similar to RING E3 ligases, U-box enzymes assemble into a functional complex with the substrate and the E2 enzyme, which directly adds ubiquitin to the target.

RING-IBR-RING (RBR) enzymes represent the newly discovered family of E3 ligases known as RING-HECT hybrid E3 ligases. The RBR E3 ligases contain a conserved catalytic region comprising RING1 and RING2 domains, and a central in-between-RING (IBR) zinc-binding domain. Mechanistically, RING1 recruits the ubiquitin-loaded E2 and then the RING2 domain, which contains the catalytic cysteine, catalyzes the transfer of ubiquitin. The IBR domain supports the catalysis when the RING2 domain lacks the catalytic cysteine residue by adopting a three-dimensional conformation similar to the RING2 domain fold.

Several regulatory mechanisms are used by E3 enzymes to modulate efficiency and substrate specificity, such as the assembly of multiprotein complexes as described above, or allosteric interaction with specific PTMs to favor substrate recognition and ubiquitination in a timely regulated manner, as exemplified by the RNF146/Iduna RING-type E3 ligase and tankyrase, a member of the poly-ADP-ribosyl transferase family [65]. RNF146 contains a specific protein domain, the WWE domain, which binds poly-ADP-ribose, the enzymatic product of poly-ADP-ribosyl transferases, in response to various stimuli [65,66,67,68,69]. The binding of tankyrase mediated-ADP-ribosylated substrates by the WWE domain of RNF146 promotes the transition of RNF146 enzyme from an inactive state to a catalytically active form which targets substrates such as the regulatory proteins Axin and 3BP2, with consequent modulation of several signaling pathways, including WNT [70].

More recently, several E3s have been reported to modify novel substrates in addition to proteins, such as lipopolysaccharide [71] and sugars [72]. For example, in the host–pathogen interaction, the host E3 ligase RNF213 modifies the lipid moiety of the lipopolysaccharide (LPS) located on the bacterial outer membrane of *Salmonella*, providing the trigger for further chain elongation that allows the recruitment of additional antibacterial effector proteins [71]. As a result, the cytosolic proliferation of *Salmonella* is impaired. In the case of sugars, an example is HOIL-1 (heme-oxidized IRP2 ubiquitin ligase-1), which associates with HOIP (HOIL-1-interacting protein) and Sharpin (Shank-associated RH domain interactor) to form a trimeric complex called the linear ubiquitin chain assembly complex (LUBAC). HOIL-1 catalyzes the attachment of ubiquitin to serine and threonine residues on target proteins and is involved in immune signaling pathways [72,73]. It has also been reported to ubiquitinate glycogen and maltoheptaose in vitro, suggesting its involvement in the modification of unbranched glycogen molecules, which may represent the triggering step for unbranched glycogen removal by cells, preventing the formation and precipitation of toxic polyglucosan aggregates [72].

### 2.2. Enzymes Involved in Ubiquitination Reversal

Ubiquitination is reversed by deubiquitinases (DUBs), which are specialized proteases that remove ubiquitin from cellular targets or cleave within ubiquitin chains to regulate ubiquitination signaling to support cellular homeostasis.

In humans, there are approximately 100 different DUBs that can be broadly grouped into seven structurally unrelated superfamilies that include the six families of cysteine proteases: (i) ubiquitin C-terminal hydrolases (UCHs); (ii) ubiquitin-specific proteases (USPs); (iii) ovarian tumor proteases (OTUs); (iv) Machado–Josephin domain proteases (MJDs, also known as Josephins); (v) the novel Ub-containing motif interacting DUB family (MINDYs); and (vi) the zinc finger-containing Ub peptidase (ZUP1); while the Jab1/Mov34/MPN+ proteases (JAMM, also known as MPN) constitute the zinc-dependent metalloproteinase family. They all interact with a common hydrophobic patch on ubiquitin [74,75]. DUBs are characterized by the presence of accessory multidomains most likely involved in protein–protein interaction or ubiquitin recognition, such as ubiquitin-binding domains (UBDs), including the zinc finger, ubiquitin-specific protease domain (ZnF-UBP domain), the ubiquitin-interacting motif (UIM) and the ubiquitin-associated domain (UBA domain) which typically binds monoubiquitin with low affinity, and the ubiquitin-like folds (UBL folds).

Mechanistically, DUBs are involved in the generation of newly synthesized ubiquitin by removing monomeric ubiquitin from multimeric precursor proteins encoded by four genes; they can also remove ubiquitin chains from post-translationally modified targets by reversing ubiquitin signaling or protein stabilization by rescuing them from proteasomal or lysosomal degradation; and they can be used to edit the form of ubiquitin modification by truncating ubiquitin chains. DUBs exhibit hydrolytic activity by cleaving ubiquitin within a chain (endo-proteolytic activity), or from one end of the chain (exo-proteolytic activity) or by removing the entire chain at once (en bloc hydrolysis) (Figure 2).

Several regulatory mechanisms ensure that DUBs are functional only at specific times and in specific subcellular localizations by modulating DUB substrate specificity and activity. For a description of the mechanisms involved in the regulation of DUBs as well as the catalytic reaction supported by the different classes of enzymes, readers are referred to very comprehensive reviews [76,77,78,79]. In this context, it is important to highlight that among the various regulatory processes, PTMs represent a tight mechanism to control the activity of DUBs. Phosphorylation of DUBs can lead to a gain of function, as in the case of OTUD5 [80,81] and A20 [82], which are phosphorylated at S177 and S381, respectively, or to a loss of function, as in the case of USP8 [83], where phosphorylation at S680 determines a decrease in catalytic activity. Other PTMs, including ubiquitination, hydroxylation, acetylation and ubiquitin-like modifiers, are also involved in the regulatory mechanisms [76].

The dysregulation of DUBs leads to the onset of several human diseases [77,84], highlighting the importance of their functions. More recently, DUBs characterized from pathogenic bacteria and viruses have been shown to play a role in evading host immune responses, leading to the onset of infectious diseases in humans and animals [38,85]. As targeting these enzymes may provide new therapeutic avenues, many efforts have been made to identify small molecules targeting DUBs for pharmacological intervention [76]. Of note, only a few of these molecules are in the clinic for cancer therapy or under investigation in clinical trials, as discussed in the next Section 5 and Section 6. An in-depth mechanistic understanding of DUBs is required for the design of new approaches.

## 3. The Ubiquitination Code

The diversity of ubiquitination-mediated cellular processes is based on the distinct conformation and topology that ubiquitin chains adopt according to the amino acid residues involved in the chemical bonds that cross-link the ubiquitin molecules (Figure 3A–D) [86,87].

In the case of monoubiquitination (Figure 3B), according to the canonical reaction mechanisms, the ubiquitin molecule is covalently bound to a specific lysine residue on the substrate via its C-terminal carboxylate group, with the formation of an isopeptide bond, as exemplified by the Lys164 residues of proliferating cell nuclear antigen (PCNA) [88]. In other cases, the number of monoubiquitinated residues varies depending on the target substrate, such as receptor tyrosine kinases (RTKs) [89,90]. Mass spectrometry analyses have shown that non-canonical ubiquitination targets other amino acid residues such as cysteine, serine, threonine or the N-terminus of the substrate [91,92]. In addition, modification of the N-terminal methionine or one of the seven lysine residues of a substrate-bound ubiquitin (Lys6, Lys11, Lys27, Lys29, Lys33, Lys48, Lys63) (Figure 3A) leads to the formation of polymer chains.

These polymers can be short, containing only two ubiquitin molecules, or long, containing more than ten units. Based on the amino acid residue involved in ubiquitin chain elongation, ubiquitin polymers can be further classified into homotypic chains, when the same residue within the ubiquitin molecule is modified during elongation, as in Met1-, Lys11-, Lys48- or Lys63-linked chains (Figure 3C); and heterotypic chains, whose assembly is based on the alternation of different linkages at successive positions of the chain, obtaining a mixed or branched topology (Figure 3D).

The different architectures of ubiquitin chains have been demonstrated in cells and their abundance changes during specific cellular processes, suggesting their different functions [93]. Indeed, monoubiquitination has been associated with proteasomal degradation, whereas multi-monoubiquitination has been linked to proteasome-independent cellular functions, including intracellular protein relocalization [94].

Instead, the assembly of long linear polyubiquitin chains triggers different cellular outcomes, such as protein recruitment and the activation of signaling cascades or proteasomal degradation, depending on the number of moieties in the polymers. The ability to assemble heterotypic ubiquitin chains gives Ub particular flexibility in its function in signaling, greatly increasing the complexity of the ubiquitination code [87]. Ubiquitin chains with mixed topology are involved in NF-kB signaling, protein trafficking, DNA repair, protein degradation or protein aggregation (Figure 3D) [95]. A deeper understanding of the crosstalk between mixed and branched ubiquitin chains, as well as the mechanisms governing the coordinated removal of these modifications, may help to understand the contribution of ubiquitination to cellular homeostasis [3,78,96]. A further layer of complexity is represented by the additional modifications of ubiquitin, including phosphorylation, acetylation, ADP-ribosylation and deamidation (Figure 3D), which lead to the establishment of diverse cellular outcomes [97], as exemplified by the ADP-ribosylation of ubiquitin [98]. The Dtx3L/Parp9 heterodimer catalyzes the NAD^+^-dependent mono-ADP-ribosylation of ubiquitin at the carboxyl terminus glycine 76, which is generally targeted by Dtx3L, a histone E3 ligase involved in DNA damage repair. ADP-ribosylation of ubiquitin interferes with Dtx3L-mediated modification, thereby preventing DNA repair [98].

Similarly, conjugation by other Ub modifiers such as ISG15, SUMO and NEDD8 further affects the structural diversity of ubiquitin chains. Readers are referred to valuable reviews to gain insight into this topic [3,98,99,100,101,102,103,104].

Ubiquitin is also recognized by many proteins via non-covalent interactions mediated by Ub-associated domains and Ub-binding domains that serve the translation of ubiquitin modification into signaling thus controlling diverse biological outcomes [97].

Since Ub interacts with a wide variety of proteins through a common surface, it is an attractive scaffold for engineering variants showing higher specificity for selective targets [105]. The application of ubiquitin engineered variants (UbVs) for developing new strategies targeting UPS is discussed in Section 6.1.

Finally, the discovery of new targets such as lipopolysaccharide [71] and sugars [72], as described in Section 2.1, suggests a higher level of complexity for ubiquitination signaling in modulating cellular outcomes.

## 4. Aberrant Ubiquitination in Cancer

The topology of the ubiquitin chains dictates the fate of the tagged substrates, marking them for proteasomal degradation, change in subcellular localization or assembly into functional complexes, as described in the paragraph above. Since E3s and DUBs are involved in the regulation of fundamental cellular outcomes, the deregulation of the signaling pathways that underpin these functions plays a role in conferring a survival advantage to cancer cells, thus linking E3s and DUBs to tumorigenesis [48,106].

The overexpression of members of the ubiquitin system, as well as the expression of their mutant proteins, has been reported in several cancer types, where they correlate with malignant progression, poor survival and prognosis [107,108,109]. Since the activity or proteolysis of several oncoproteins and tumor-suppressor proteins is impaired by the deregulation of the ubiquitin system in cancer [48], numerous biological processes are altered, including the cell cycle [15,51,110], cell invasion and metastasis [110,111], cancer metabolism [112,113], cancer stem cell stemness [113,114], immunological tumor microenvironment [113], immune evasion [51], tumor-promoting inflammation [51] and evasion from apoptosis [51,110].

Accumulating evidence also points to a proteasome-independent role for ubiquitination in oncogenic pathways through the monoubiquitination and polyubiquitination of various substrates [115,116,117,118]. For example, the monoubiquitination of histone H2B at the Lys120 position can modulate many key cellular mechanisms, including the DNA damage response and transcription, by physically relaxing chromatin to allow access of DNA repair proteins and transcription factors [115]. Reduced levels of this ubiquitination lead to abnormal transcriptional elongation of estrogen receptor α-targeted genes. Furthermore, reduced levels of histone modification H2B have been described during breast tumorigenesis and correlate with the different stages of oncogenesis [115]. Altered H2B ubiquitination has also been reported in gastric, parathyroid and colorectal tumors. In inflammatory bowel cancer, mutations in RNF20 correlate with decreased levels of H2B ubiquitination, and RNF20-deficient mice are prone to inflammatory bowel cancer. This suggests that H2B ubiquitination may be a tumor suppressor and may represent a novel druggable target in cancer therapy. Monoubiquitination of H2B can be restored by the inhibition of the deubiquitinases USP12, USP44 and USP7 [115]. Defects in deubiquitination have effects on the NF-kB, AKT and BRCA signaling pathways [119]. For a more comprehensive description of the role of the ubiquitination system in cancer development and progression, readers are referred to specific reviews [15,51,110,111,112,113,114,120,121,122,123,124,125,126,127].

Interestingly, ubiquitination is also involved in modulating the homeostasis of cellular organelles whose dysfunction plays a role in tumorigenesis. The Golgi Complex (GC) is an example, as functional and structural alterations of the GC [128,129,130] and the dysregulation of GC-centered signaling cascades [131] have been observed in cancer. There is evidence that a small number of E3s localize to the GC, where they mediate the ubiquitination of resident proteins, thereby regulating organelle structural integrity and vesicle trafficking [132]. For example, the CUL7^FBXW8^ complex regulates the turnover of GRASP65. In mammalian brain neurons, the CUL7^FBXW8^ complex induces the ubiquitination and degradation of GRASP65 to control GC morphology and dendrite patterning by regulating secretory trafficking [133]. In mammalian dividing cells, GRASP65 is involved not only in maintaining the GC structure but also in regulating the Golgi mitotic checkpoint through JNK2-mediated phosphorylation of GRASP65 [134]. Therefore, similar to that observed in mammalian brain neurons, it can be speculated that further PTMs of GRASP65, including ubiquitination, affect the stability/function of GRASP65 with effects on GC structure and cell cycle progression. This is not surprising as other PTMs, such as ADP-ribosylation, have been shown to play a role in maintaining GC structure and function during stress responses [67]. Therefore, abnormal ubiquitination of GC proteins may also play a role in carcinogenesis.

Therefore, E3s and DUBs, which have been identified as novel druggable targets, represent an opportunity for the development of anticancer therapeutic approaches, as discussed later in Section 6.

## 5. Targeting Ubiquitination as a Therapeutic Approach to Cancer Treatment

The discovery that ubiquitin signaling plays degradative and non-degradative roles in many biological processes and, more interestingly, that its deregulation plays a role in oncogenesis, has stimulated interest in targeting ubiquitination as a novel therapeutic approach. Over the last decades, several natural or synthetic compounds have been investigated for their ability to target the ubiquitination machinery and interfere with tumor progression. This section focuses on the ubiquitination-targeting drugs currently used in the clinic for cancer treatment and the main results obtained in clinical trials to date.

### 5.1. Food and Drug Administration (FDA)-Approved Molecules in the Clinic

Table 1 lists the compounds that target components of the ubiquitin–proteasome system (including E3 ligases, proteasome and DUBs) that have been approved by the FDA for the treatment of selected cancers.

Bendamustine inhibits the E3 ligase linear ubiquitin assembly complex (LUBAC). The LUBAC assembles ubiquitin chains in which ubiquitin is linked to the amino-terminal methionine at position 1 (M1) of another ubiquitin (referred to as ‘linear’ or ‘M1-linked’ ubiquitin chains). LUBAC and M1-linked polyubiquitin regulate multiple signaling pathways, the best characterized of which is the NF-kB pathway, which controls immune responses, inflammation and cell survival [135].

Thalidomide, lenalidomide and pomalidomide are immunomodulatory drugs with anticancer activity against multiple myeloma by inhibiting the cereblon protein (CRBN). CRBN is a substrate receptor of the cullin 4-really interesting new gene (RING) E3 ubiquitin ligase complex CRL4^CRBN^. Targeting CRBN promotes ubiquitination-mediated proteasomal degradation of neosubstrates, including MEIS2, IKZF1, IKZF3, CD147, casein kinase 1 α, MCT1, ZMYM2 and ZMYM2-FGFR1, resulting in the inhibition of cell proliferation and induction of apoptosis [136,137,138,139,140].

Bortezomib and ixazomib are reversible inhibitors of the proteasome, while carfilzomib is an irreversible proteasome inhibitor. All three inhibitors target the chymotrypsin-like β5 catalytic subunit of the proteasome, thereby blocking the degradation of multiple proteins, including p53, cyclins, Cdks and/or Cdk inhibitors, IκB, and thus inhibiting tumor growth [141]. These drugs have been approved for the treatment of multiple myeloma and mantle cell lymphoma. However, several clinical trials are currently investigating the therapeutic efficacy of these proteasome inhibitors in other tumor types, including hematological and solid malignancies, both as monotherapy and in combination with other drugs (https://clinicaltrials.gov).

Mitoxantrone is a synthetic anthracenedione approved by the FDA for the treatment of several tumor types, including acute leukemia, lymphoma, prostate and breast cancer and multiple sclerosis. Mitoxantrone has been shown to have multiple cellular targets, including DNA topoisomerase II [142], the serine/threonine kinase Pim1 [143], focal adhesion kinase [144], USP11 [145] and USP15 [146], providing a rationale for its therapeutic efficacy in the treatment of multiple human diseases.

An antineoplastic and immunosuppressive agent, 6-Mercaptopurine (6-MP), and 6-thioguanine (6-TG), an antimetabolite, are natural purine analogs approved by the FDA for the treatment of acute lymphoblastic leukemia and inflammatory bowel disease. In cells, 6-MP and 6-TG are converted to nucleotides that inhibit both de novo biosynthesis and the interconversion of normal purines and nucleic acid metabolism [147]. The finding that 6-MP and 6-TG inhibit several DUBs, such as USP14 [148], USP2a [149,150,151], involved in tumorigenesis, and PLpro [148], suggests that these molecules could affect cancer through additional molecular mechanisms of action.

### 5.2. Major Achievements in Clinical Trials Targeting Ubiquitination

In the last decades, a large number of drugs/molecules targeting the ubiquitination machinery have entered clinical trials against solid and hematological malignancies (for a review, the reader is referred to [51,110,122,125,126,152,153]) and several trials have been conducted and are currently ongoing to assess their safety, tolerability and therapeutic activity in cancer (https://clinicaltrials.gov). Within this growing number of trials, only a few have been completed and results are currently available (Table 2).

As shown in Table 2, most UPS inhibitors have been evaluated in combination with chemotherapeutic agents currently used in the clinic. They showed therapeutic efficacy mainly against hematological malignancies, with the disappearance of all lesions in 50% of participants in a combinatorial treatment regimen using pevonedistat and azacitidine (NCT01814826). On the other hand, results were disappointing in the treatment of solid tumors, where the best complete response in combinatorial strategies was 25% (NCT01617668). It is worth noting that the route of drug administration influences the therapeutic efficacy of treatment (NCT01814826). In addition, patients recruited in all clinical trials experienced serious adverse events, including general, vascular, gastrointestinal, cardiac, respiratory and thoracic, hematological and lymphatic disorders. For more information concerning the treatment-related side effects, the readers are referred to the link https://clinicaltrials.gov.

Of note, a recent study showed that disulfiram, an inhibitor of aldehyde dehydrogenase approved by the FDA for the treatment of alcoholism, targets the cysteine residue of the catalytic domain of USP2 and USP21 [151]. As a result, several clinical trials have been conducted to expand disulfiram’s indications for the treatment of various cancers, including pancreatic cancer, breast cancer, non-small cell lung cancer, glioblastoma and glioma (https://clinicaltrials.gov). Of these, completed clinical trials with published results are listed in Table 2.

## 6. Developing Strategies to Target Ubiquitination in Cancer Therapy

Several components of the UPS (such as E1, E2, E3, proteasome and DUBs) have been investigated in preclinical and clinical studies as therapeutic targets in cancer treatment [51,125,126,127,153]. The therapeutic strategies developed so far are mainly based on: (i) the possibility of manipulating the ubiquitination machinery to regulate specific proteins involved in cancer development and progression, i.e., oncoproteins and oncosuppressors; and (ii) exploiting the vulnerability of cancer cells to develop synthetic lethality approaches. The main achievements are described below and summarized in Figure 4.

### 6.1. Targeting Ubiquitin–Proteasome System to Modulate Tumor-Suppressor Proteins and Oncoproteins

Tumor-suppressor proteins, such as PTEN and p53, are often mutated in several types of cancer. The selective and specific inhibition or activation of UPS components, which are responsible for regulating the stability, activity, subcellular localization or protein–protein interaction of tumor suppressors, is a promising approach for restoring their functionality.

The lipid phosphatase PTEN acts as a tumor suppressor by interfering with the oncogenic activity of the AKT signaling pathway. Several E3s (including Nedd4-1, WWP1, WWP2, XIAP and CHIP) and DUBs (such as USP7, USP13 and OTUD3) modulate the ubiquitination state of PTEN, which in turn regulates its subcellular localization, stability and activity, thereby regulating AKT signaling. Therefore, targeting PTEN ubiquitination states by pharmacologically targeting E3 ligases or DUBs leads to the inhibition of the AKT pathway and consequent prevention of tumor growth [113,154]. In the same line of evidence, AKT stability, activity and localization are modulated by specific E3s and DUBs. AKT activation depends on K63-linked ubiquitination-mediated localization at the plasma membrane and is essential for cell proliferation. TRAF6, SKP2 and Nedd4 are some of the E3s responsible for AKT K63-linked ubiquitination and activation. In contrast, K48-linked ubiquitination of AKT regulates AKT stability. In addition, AKT ubiquitination mediated by CHIP, MULAN and TTC3 E3s promotes its degradation. Similarly, DUBs regulate AKT degradation, localization, activation and protein–protein interactions. For example, the CYLD deubiquitinase removes AKT K63-linked ubiquitination upon growth factor stimulation, resulting in BRCA1- or TTC3-mediated K48-linked polyubiquitination, thus leading to AKT degradation. Therefore, targeting AKT E3s and DUBs represents a therapeutic avenue in cancer treatment [113,154]. Among the molecules targeting the E3s and DUBs of the PTEN/AKT pathway [154], indole-3-carbinol, a natural inhibitor of NEDD4 and WWP1, induces PTEN upregulation and has been investigated in clinical trials for the treatment of breast (NCT00033345) and prostate (NCT00607932) cancer, although the results of these trials have not been reported on ClinicalTrials.gov.

p53 is another example of a tumor-suppressor protein and is one of the major tumor suppressors downregulated in cancer. p53 stability, subcellular localization and activity are regulated by its ubiquitination state, which is controlled by the activities of several E3s and DUBs [113]. MDM2, which is overexpressed in several cancer types, is the major E3 responsible for p53 ubiquitination and proteasomal degradation in cancer. Targeting both the MDM2-p53 interaction and MDM2 activity with small molecule inhibitors is a therapeutic approach being investigated in clinical trials for solid and hematological malignancies [51,155]. In particular, inhibitors of MDM2–p53 interaction that have entered clinical trials include RG7388 and RG7112, while APG-115 is an inhibitor of MDM2 activity that is currently in clinical trials (https://clinicaltrials.gov). Another promising approach to stabilizing p53 in cancer cells consists of promoting the MDM2 proteasomal degradation through the Ub variant (UbV)-mediated inhibition of USP7 and USP2a deubiquitinases, known to be involved in stabilizing MDM2. The rationale of this strategy relies on several properties, including the presence of Ub-binding sites in the UPS proteins, the ability of Ub to engage in low-affinity but specific interactions with UPS components and the possibility of genetically engineering the ubiquitin molecules. Over the past few years, UbVs have been obtained and screened against multiple components of UPS, such as E2 conjugating enzymes, E3 ligases and DUBs [156]. UbVs show high affinity and selectivity in targeting specific UPS protein and they can act as inhibitors, activators or modulators of targeted proteins. As exemplars, UbVs showing high affinity and selectivity for USP7 and USP2a have been identified. The intracellular expression of these UbVs in cancer cells inhibits endogenous USP7 and USP2a, thus promoting the proteasomal degradation of MDM2 and, consequently, p53 stabilization [157,158,159]. Although the UbVs targeting UPS components have demonstrated their anticancer efficacy in cellular models [156], their potential use as therapeutics in cancer treatment awaits further validation.

Similar to the tumor-suppressor proteins, E3 ligases and DUBs play a role in modulating the stability, activity, subcellular localization or protein–protein interaction of oncoproteins. The transcription factor c-Myc is one of the most frequently activated oncoproteins in human neoplasia. It regulates cell proliferation, metabolism and metastasis by controlling multiple signaling pathways. c-Myc protein turnover is mediated by the activities of E3s and DUBs. The E3s SKP2, Fbw7, β-TrCP1, CHIP and FBXO32 promote c-Myc ubiquitination-mediated proteasomal degradation, thereby inhibiting tumorigenesis. In contrast, USP37- and USP36-mediated deubiquitination can promote tumorigenesis by stabilizing c-Myc [113,160]. Therefore, the activation of c-Myc E3s or inhibition of c-Myc DUBs may have therapeutic implications in cancer. WB100 and tanshinone IIA are two molecules that induce c-Myc degradation by targeting the E3s CHIP and Fbxw7, respectively. These molecules have entered clinical trials for the treatment of solid tumors and hematological malignancies (WB100 clinical trial: NCT05100251; tanshinone IIA clinical trials: NCT01452477, NCT02200978) [160].

An interesting strategy to promote ubiquitination-mediated degradation of a specific undruggable oncoprotein is the proteolysis targeting chimeras (PROTACs) [161]. PROTACs are bivalent molecules in which a ligand specific for a protein of interest is fused via a linker to another ligand specific for an E3. These molecules recruit the E3 to the protein target, thereby stimulating its proximity-induced ubiquitination and degradation by the proteasome. PROTACs have several advantages over classical inhibitors of ubiquitination, including high specificity, the potential to target undruggable proteins, therapeutic efficacy at sub-stoichiometric concentrations, thereby reducing toxicity, the ability to overcome inhibitor-induced resistance by targeting mutant proteins, and recycling after the target protein has been ubiquitinated. ARV-110 and ARV-471 are two PROTACs targeting the androgen receptor and the estrogen receptor, respectively, that have entered clinical trials for the treatment of prostate and breast cancer (ARV-110 clinical trials: NCT03888612 and NCT05177042; ARV-471 clinical trials: NCT05463952, NCT04072952, NCT05573555, NCT05501769, NCT05548127, NCT05549505, NCT05732428, NCT05909397, NCT05654623, NCT01042379). Recently, Murgai and collaborators [162] developed the first PROTAC-based DUB degrader that induces the degradation of USP7, a negative regulator of p53, that is overexpressed in several cancers [163]. In addition, the PROTAC-based MDM2 degrader MD-224 effectively induces rapid degradation of MDM2 and achieves tumor regression in an in vivo leukemia xenograft tumor model [164].

### 6.2. Targeting Ubiquitin–Proteasome System to Induce Synthetic Lethality

The understanding of the molecular mechanisms underlying the UPS- and DUB-mediated regulation of biological processes and the molecular characterization of the tumors pave the way for the development of novel therapeutic approaches based on the induction of synthetic lethality. The use of PARP inhibitors for the therapeutic treatment of tumors defective in homologous recombination repair (HR) due to BRCA1/2 mutations (including breast, ovarian and metastatic castration-resistant prostate cancer) has provided the first clinical proof of concept for synthetic lethality [165]. PARP inhibitors trap PARP1 in an inactive cytotoxic conformation, resulting in increased DNA replication stress, DNA damage, cell cycle arrest and cell death. To date, the use of PARP inhibitors has been limited to the treatment of BRCA1-deficient cancers. In addition, resistance to PARP inhibitors often occurs in the clinic, highlighting the need to develop new therapeutic avenues. The identification of novel DNA damage vulnerabilities could lead to the use of PARP inhibitors in the treatment of BRCA1-proficient cancers and provide new therapeutic avenues to overcome PARP inhibitor resistance. The identification of USP1 deubiquitinase as a key player in replication fork protection in BRCA1-deficient cells underlies the synthetic lethal interaction between both USP1 and BRCA1 deficiency [166,167]. BRCA1-deficient cells exhibit a deficiency in HD repair, with a reduced ability to protect stalled replication forks and to repair DNA double-strand breaks. These defects are exacerbated by the loss of USP1. USP1, which is localized at the replication fork, deubiquitinates PCNA and thus stabilizes the fork. In the absence of USP1, mono- and polyubiquitinated PCNA accumulates at the replication fork and drives the recruitment of translesion synthesis polymerases that introduce DNA gaps and mismatches, leading to fork destabilization, decreased DNA synthesis, S-phase arrest and cell death in BRCA1-mutant cancer cells [166,167]. Based on this mechanism of action, combinatorial therapy with both USP1 and PARP inhibitors may have clinical efficacy in BRCA1-deficient cancers with acquired resistance to PARP inhibitors. The USP1 inhibitors KSQ-4279 and ISM3091 are currently undergoing Phase I clinical trials to evaluate their safety, tolerability, clinical activity and pharmacokinetics in the treatment of advanced solid tumors (NCT05240898 and NCT05932862 respectively). The data may pave the way for the development of synthetic lethal regimens in combination with PARP inhibitors. Recent evidence shows that the deubiquitinase TRABID is overexpressed in prostate cancer, contributing to HR repair deficiency [168]. TRABID binds to and catalyzes the deubiquitination of polyubiquitinated 53BP, which mediates the inactivation of DNA end resection thus promoting non-homologous end joining (NHEJ) over HR repair. TRABID-mediated deubiquitination of 53BP1 results in prolonged retention of 53BP1 at the double-strand break sites, thereby inhibiting HR. Therefore, TRABID overexpression-mediated HR deficiency enables the potential therapeutic use of PARP inhibitors in the treatment of prostate cancer in a synthetic lethality setting [168]. TRIP12 is an E3 that binds to and catalyzes the polyubiquitination of PARP1, leading to its proteasomal degradation and preventing supraphysiological PARP1 accumulation. PARP1 accumulation mediated by TRIP12 dysfunction sensitizes BRCA-proficient cancer cells to PARP inhibitors through increased PARP1 trapping [169]. Therefore, the identification of TRIP12-deficient tumors and the development of TRIP12 inhibitors may allow the use of PARP inhibitors in the treatment of BRCA1 proficient cancers.

## 7. Conclusions and Perspectives

This perspective has described the complexity of ubiquitination in the maintenance of cellular homeostasis, focusing on the effects of deregulation of ubiquitination in cancer. The discovery of new antitumor therapeutic interventions based on the selective inhibition of the ubiquitination–proteasome axis remains therefore a primary goal to be achieved. The severe side effects and drug resistance experienced by cancer patients treated with FDA-approved proteasome inhibitors (Table 1) clearly highlight the need for new approaches with valuable therapeutic efficacy. In order to meet these needs, in the last two decades numerous studies have been undertaken, and a huge number of molecules targeting UPS components have entered clinical trials as single agents or in combination with FDA-approved chemotherapeutic agents (https://clinicaltrials.gov). Among the countless clinical trials, a few studies have been completed so far, as shown in Table 2. On the other hand, the majority of them has not been completed, due, for example, to the unavailability of fundings, and so the results are partial and not fully informative, resulting in a lack of therapeutic approaches targeting UPS in the treatment of cancer, particularly solid tumors. Hence, despite the promising results obtained in preclinical studies on cellular and animal models, the targeting of UPS remains challenging due to the complexity of the ubiquitination code, the regulatory networks of UPS enzymes and tumor biology. Consequently, in order to render ubiquitination targeting a viable approach in clinic, several challenges need to be met: (1) a deeper understanding of the ubiquitination code, (2) achieving the specific targeting of proteins involved in UPS, (3) dissecting the molecular mechanisms that regulate/dysregulate the activity of UPS components, (4) investigating the role of UPS proteins in the context of each tumor, (5) analyzing the genetic alterations and the expression profile of UPS components in the specific tumoral context, (6) developing inhibitors/agonists of UPS enzymes with more favorable pharmacokinetic and pharmacodynamics profiles, and (7) implementing more efficient targeted drug-delivery systems. Addressing these items will allow the translation of UPS targeting approaches to the clinic as well as designing personalized medicine strategies in combinatorial and/or synthetic lethality treatment regimens.

As it concerns the improvement of the specificity and selectivity of targeting, one of the most promising approaches relies on targeting the E2–E3 complex [48,113] or E2s [170], since the E2–E3 interaction confers high specificity and selectivity of the response by affecting specific Ub bonds. Similarly, targeting the substrate binding domains of E3s represents a valuable opportunity to explore. In particular, inhibiting the interaction between a given E3 and its target increases specificity and reduces off-target and side effects, most likely due to the limited number of cellular events affected. The inhibition of the MDM2-p53 interaction discussed in Section 6.1 is an example of this approach.

As protein–protein interaction also influences the specificity and selectivity of E3s, a deeper understanding of the three-dimensional structure of E3 enzymes by crystallography and of the interaction sites by cryo-electron microscopy, to mention the most relevant approaches, will provide new clues for the development of novel inhibition strategies. Following the same line of evidence, the exploitation of engineered ubiquitin variants (UbVs) will pave the way to improve target specificity and selectivity. In fact, in the last few years, several studies have demonstrated that UbVs that tightly bind to UPS–related enzymes (such as E2s, E3s, and DUBs) may serve as valuable tools to modulate the function of these enzymes [156], as exemplified by the UbVs which inhibit USP7 and USP2a deubiquitinases described in Section 6.1.

Similarly, with the aim of increasing substrate specificity, PROTACs [161], previously described in Section 6.1, and molecular glues [171] constitute a useful approach to promote the ubiquitination-mediated degradation of a specific protein. Immunomodulatory drugs such as thalidomide, lenalidomide and pomalidomide, described in Section 5.1, are prominent examples of molecular glues targeting the neo-substrates involved in promoting tumorigenesis. However, molecular glues are also a promising tool for enhancing the native protein–protein interaction between an oncogenic protein and its cognate E3 ligase, which is otherwise weakened in the diseased state. An example of this is the molecular glue NRX-2663, which is able to enhance the protein–protein interaction between β-catenin and its cognate E3 ligase SCF^β-TrCP^ [172]. The phosphorylation of β-catenin at Ser33 and Ser37 mediates its interaction with SCF^β-TrCP^, leading to ubiquitination-mediated degradation of β-catenin. In many cancers, this interaction is significantly attenuated due to mutations or decreased phosphorylation of β-catenin. The NRX-2663 enhancer potentiates the ubiquitination of mutant β-catenin by SCF^β-TrCP^ in vitro, although it is less potent in vivo [172].

There are important advantages in using the molecular glue approach: (i) binding to the protein of interest creates a new surface available for interaction with the desired E3 partner; (ii) the small size allows molecular glues to enter cells and tissues with greater efficiency compared to PROTACs. At the same time, both PROTAC and molecular glue-based approaches are capable of degrading tumor-specific proteins via tumor-overexpressed E3s. Thus, the implementation of -omics technologies that increase the knowledge of proteins and signaling pathways deregulated in tumors, would allow the identification of tumor-specific and selective molecular targets useful for the design of novel PROTAC- and molecular glue-based therapeutic approaches.

Another limitation associated with UPS inhibitors is the adverse side effects caused by non-specific drug distribution, which kills both healthy and malignant cells, resulting in systemic cytotoxicity. Targeted drug delivery offers a way around this problem. In fact, it offers several advantages, including the reduction in side effects, the administration of a lower drug dosage and the specific increase in drug concentration on tumors, thus achieving greater overall therapeutic benefit. To this end, the functionalization of nanoparticles with targeting ligands (including small molecules, peptides, proteins, antibodies, nanobodies and aptamers [173,174,175,176]) promotes selective binding to the tumor-specific complementary receptor (such as proteins, lipids or carbohydrates), resulting in drug-specific internalization in tumor cells. Therefore, in order to implement new targeted drug-delivery strategies, further -omics studies should be carried out to characterize the tumor surface expression signature in detail.

Another promising approach to targeted drug delivery is the use of cell membrane-coated nanoparticles (CNPs) [177], an emerging class of nanocarriers. CNPs consist of a synthetic core containing the anticancer drug, covered by an outer layer of naturally derived cell membranes, which guide the interaction of the CNPs with the surrounding cells, proteins and other biological substrates. The functionalization of CNPs with specific cell membranes drives the delivery and accumulation of CNPs at disease sites [177]. In addition to the above strategies, exosomes might be used as an alternative vehicle for the delivery of ubiquitination-targeting drugs. Similar to nanoparticles, exosomes can be engineered to express functional molecules (including peptides, antibodies, aptamers, glycans) on their surface for targeting to a specific disease site [178,179]. Although the use of CNPs and exosomes as anticancer drug-delivery vehicles has been successful in preclinical studies, translation to the clinic requires further technical improvements, including the development of efficient large-scale production and purification methods. Microenvironment-responsive drug-delivery systems (including pH-, redox- and hypoxia-responsive systems) based on nanoparticles and exosomes may serve as delivery tools to guide the release of inhibitors/agonists of UPS components in the cancer-specific microenvironment [179,180,181,182]. Therefore, a deeper understanding of the biology of the tumor microenvironment is a fundamental prerequisite for the development of novel targeted drug-delivery systems.

Recently, approaches for the targeted delivery of inhibitors of DUBs have been developed and evaluated for anticancer efficacy in preclinical studies. Shang and co-workers demonstrate that nanoparticles encapsulating the USP1 inhibitor and cisplatin efficiently impair tumor growth in a patient-derived xenograft mouse model of liver cancer [183]. Similarly, a nanoparticle formulation of an inhibitor of UCH-L1 DUB suppresses the metastatic properties of advanced oral squamous cell carcinoma cells in vitro [184]. Although these strategies have demonstrated anticancer activity in preclinical studies, their translation into clinical practice requires further efforts.

Lastly, the development of combinatorial and synthetic lethality strategies targeting UPS represents one of the most challenging approaches to improve anticancer therapeutic efficacy [185,186]. The clinical trials listed in Table 2 demonstrate the higher therapeutic efficacy of dual strategies compared to UPS inhibitors as single agents. Several combinatorial therapies that are currently under investigation [152] combine FDA-approved proteasome inhibitors with chemotherapeutic agents to overcome resistance to these inhibitors and enhance their antitumor efficacy. Therefore, the identification of synergistic combinations to evaluate for cancer treatment is of great importance. To this end, computational methods based on transcriptomics and protein interaction networks [187], as well as drug repurposing opportunities arising from artificial intelligence-based analyses [188], represent valuable tools to limit the number of drug combinations for screening.

Over the past two decades, anticancer treatment strategies based on the synthetic lethal interactions of DNA damage-response genes have been an area of intense research activity. In this context, the preclinical studies described in Section 6.2 demonstrate the anticancer efficacy of combining ubiquitination-mediated DNA damage vulnerabilities with PARP inhibitors. Notably, these studies pave the way for novel therapeutic avenues and highlight the need to perform CRISPR/Cas9- or RNAi- or shRNA-based whole genome screening in combination with UPS targeting. The experimental data, combined with genomic and transcriptomic data from public databases, should support the prediction of new potential synthetic lethal interactions to exploit alternative anticancer therapeutic approaches [189].

Even though efforts have been made to optimize the formulation of UPS inhibitors to increase specificity, delivery and retention time in tumors, the development of potent inhibitors still remains a challenging issue in drug discovery.

More recently, the discovery of new substrates targeted by ubiquitination, such as sugars in addition to proteins, is broadening the interest in this PTM as a master regulator of a multitude of cellular outcomes. Indeed, the involvement of HOIL-1 in the modification of unbranched glycogen molecules to prevent the formation of polyglucosan aggregates [72] paves the way for novel therapeutic approaches to treat other pathologies of public health interest, such as neurodegenerative diseases.

## Figures and Tables

**Figure 1 cells-13-00029-f001:**
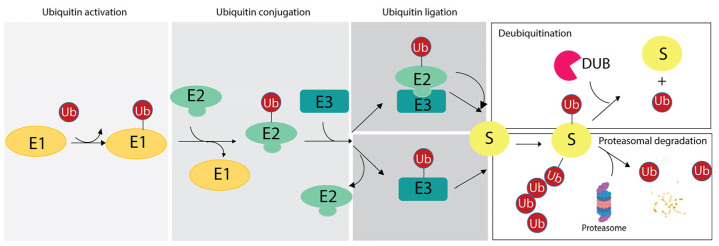
Ubiquitination reaction and fate of targeted substrates based on the ubiquitin (Ub) chains. The scheme illustrates the enzymatic reaction involving the E1 ubiquitin-activating enzyme (E1), the E2 ubiquitin-conjugating enzyme (E2) and the E3 ubiquitin-ligases (E3). As two distinct mechanisms of ubiquitin ligation may occur, two examples are reported herein. Then, based on the number of ubiquitin residues and linkage specificity, targeted substrate may undergo the DUB-mediated removal of ubiquitin (upper panel on the right) or it can be degraded by proteasome with the release of free ubiquitin moieties (lower panel on the right).

**Figure 2 cells-13-00029-f002:**
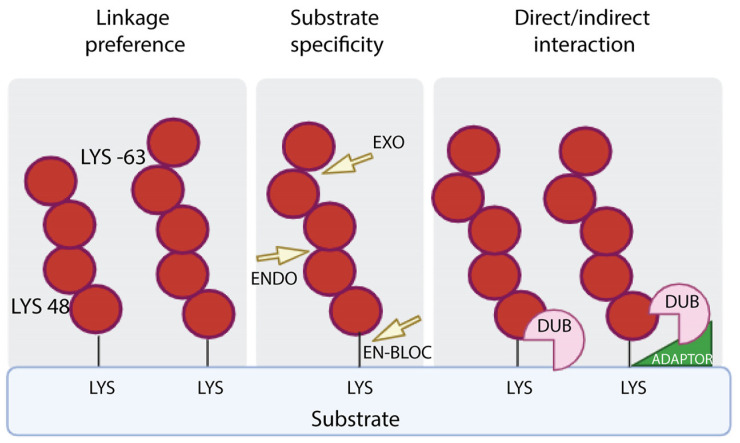
DUB specificity. DUBs perform the removal of ubiquitin from targets on the basis of the specificity of linkage (**left** panel), the mode of action, which refers to removal of the distal ubiquitin (EXO) or internal ubiquitin along the chain (ENDO) or the proximal ubiquitin (EN-BLOC) (**central** panel), the interaction with targeted proteins with or without the assistance of adaptors (**right** panel).

**Figure 3 cells-13-00029-f003:**
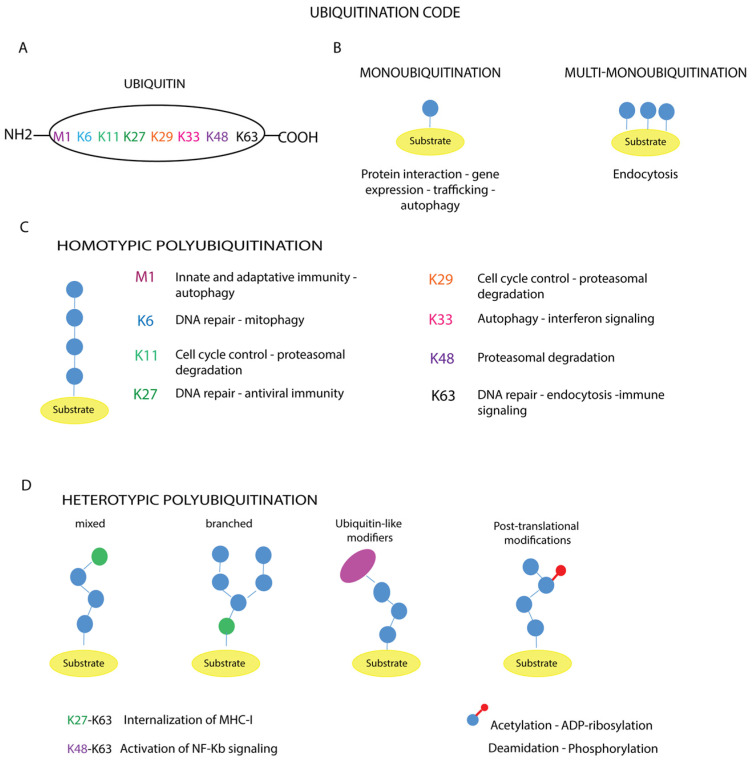
Ubiquitination code. (**A**) The amino acid residues of ubiquitin involved in the modification are reported. The different types of ubiquitin chains with topologies are shown (**B**–**D**). The schematic representation illustrates the linkage specificity associated with the corresponding cellular functions: (**B**) monoubiquitination and multi-monoubiquitination, (**C**) homotypic polyubiquitination, (**D**) heterotypic polyubiquitination.

**Figure 4 cells-13-00029-f004:**
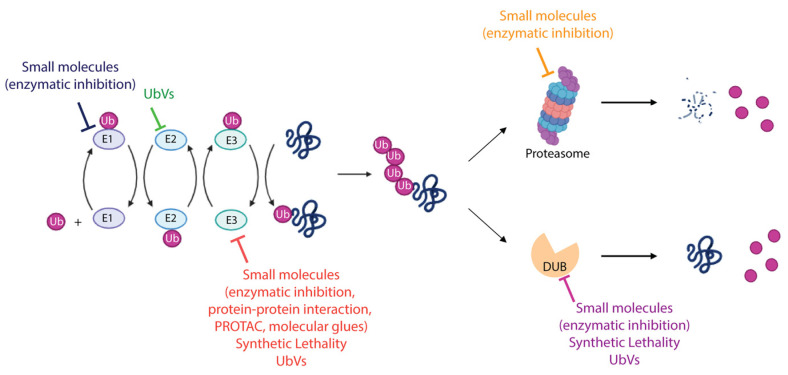
Therapeutic modalities targeting ubiquitin–proteasome system. The schematic representation summarizes the different strategies adopted to target the enzymes involved in the UPS. FDA-approved molecules, compounds entered in clinical trials, both cited as small molecules, as well as recently discovered approaches including synthetic lethality and UbV-mediated modulation of UPS proteins are indicated. More details are provided in the text. Created with BioRender.com, accessed 10 December 2023.

**Table 1 cells-13-00029-t001:** FDA-approved drugs targeting ubiquitin–proteasome system in anticancer therapies.

Drug	Protein Target	Cancer Treatment
Bendamustine	E3 ligase linear ubiquitin assembly complex (LUBAC)	Multiple myeloma, chronic leukemia, rituximab/refractory follicular and low-grade lymphoma, ovarian cancer
Thalidomide	E3 ligase cereblon (CRBN)	Myeloma
Lenalidomide	E3 ligase cereblon (CRBN)	Myeloma
Pomalidomide	E3 ligase cereblon (CRBN)	Myeloma
Bortezomib (Velcade)	26S Proteasome (β5)	Multiple myeloma, mantle cell lymphoma
Carfilzomib (Kyprolis)	26S Proteasome (β5)	Multiple myeloma, mantle cell lymphoma
Ixazomib (MLN2238, Ninlaro)	20S Proteasome (β5)	Multiple myeloma
Mitoxantrone	USP11, USP15	Acute nonlymphocytic leukemia, prostate cancer
6MP, 6TG	USP14, USP2a, PLpro	Acute lymphocytic leukemia

**Table 2 cells-13-00029-t002:** Drugs entered clinical trials targeting ubiquitin–proteasome system in anticancer therapies. An overview of the clinical trials completed with posted results on ClinicalTrials.gov website is reported. Data from https://clinicaltrials.gov (accessed on 30 October 2023).

**E1, E2, and E3 Enzyme Inhibitors**						
**Drug**	**Target**	**ClinicalTrials.gov Identifier**	**Treatment**	**Phase**	**Cancer Type**	**Results ***
Tak-243 (MLN7243)	Ubiquitin-activating enzyme (UAE)	NCT02045095	MLN7243	Phase I	Advanced malignant solid tumors	1 mg: CR = 0%, PR = 33%, SD = 67%, PD = 0% 2 mg: CR = 0%, PR = 0%, SD = 75%, PD = 25% 4 mg: CR = 0%, PR = 0%, SD = 100%, PD = 0% 8 mg: CR = 0%, PR = 0%, SD = 0%, PD = 100% 12 mg: CR = 0%, PR = 0%, SD = 50%, PD = 50% 18 mg: CR = 0%, PR = 0%, SD = 60%, PD = 40%
MLN4924 (Pevonedistat)	NAE (E1)	NCT01862328	MLN4924 Paclitaxel Gemcitabine Docetaxel Carboplatin	Phase I	Solid tumors	MLN4924 + Paclitaxel + Carboplatin: CR = 18% MLN4924 + Docetaxel: PR = 19% MLN4924 + Carboplatin: PR = 17% MLN4924 + Paclitaxel + Carboplatin: PR = 36%–40%
		NCT02122770	MLN4924 Fluconazole Itraconazole Docetaxel Carboplatin Paclitaxel	Phase I	Advanced solid tumors	MLN4924 + Docetaxel: OR = 10.5% MLN4924 + Carboplatin or Paclitaxel: OR = 22.2%
		NCT03057366	Pevonedistat [14C]-Pevonedistat Docetaxel Carboplatin Paclitaxel	Phase I	Advanced solid tumors	Pevonedistat + Paclitaxel + Carboplatin: CR = 0%, PR = 0%, SD = 40%, PD = 60% Pevonedistat + Docetaxel: CR = 0%, PR = 0%, SD = 0%, PD = 100%
		NCT03330106	Pevonedistat Docetaxel Carboplatin Paclitaxel	Phase I	Advanced solid tumors	Pevonedistat + Docetaxel: OR = 9.1% Pevonedistat + Carboplatin + Paclitaxel: OR = 8.3%
		NCT03486314	Pevonedistat Rifampin Docetaxel Carboplatin Paclitaxel	Phase I	Advanced solid tumors	Pevonedistat + Docetaxel: CR = 0%, PR = 33.3%, SD = 33.3%, PD = 33.3% Pevonedistat + Carboplatin + Paclitaxel: CR = 0%, PR = 14.3%, SD = 42.9%, PD = 42.9%
		NCT01814826	MLN4924 Azacitidine	Phase I	Acute myelogenous leukemia	MLN4924 (20 mg/m^2^) + Azacitidine (IV administration): CR = 43%, PR = 14%, CRi = 4% MLN4924 (20 mg/m^2^) + Azacitidine (SB administration): CR = 33%, PR = 13%, CRi = 13% MLN4924 (30 mg/m^2^) + Azacitidine (IV administration): CR = 50%, PR = 0%, CRi = 0%
		NCT02610777	Azacitidine Pevonedistat	Phase II	Myelodysplastic syndromes leukemia, myelomonocytic, chronic leukemia, myeloid, acute	Azacitidine: CR = 36%, OR = 45% Azacitidine + Pevonedistat: CR = 45%, OR = 51%
LCL161	RING-type E3 IAPs	NCT01617668	LCL161 paclitaxel	Phase II	Triple negative breast cancer	LCL161 + paclitaxel (gene expression signature positive): CR = 24.9% Paclitaxel (gene expression signature positive): CR = 23.4% LCL161 + paclitaxel (gene expression signature negative): CR = 6.9% Paclitaxel (gene expression signature negative): CR = 9.1%
		NCT01955434	LCL161 Cyclophosphamide	Phase II	Recurrent plasma cell myeloma refractory plasma cell myeloma	No results available on the CR, PR, SD, and OR
TL-32711 (birinapant)	RING-type E3 IAPs	NCT01188499	Birinapant, Carboplatin/Paclitaxel Irinotecan Docetaxel Gemcitabine Liposomal Doxorubicin	Phase I Phase II	Advanced or metastatic solid tumors	Carboplatin/paclitaxel + TL32711 Irinotecan + TL32711 Docetaxel + TL32711 Gemcitabine + TL32711 Liposomal doxorubicin percent patients overall across all five arms demonstrating CR or PR = 10%
AT-406 (DEBIO1143)	IAPs	NCT04122625	Debio1143 Nivolumab	Phase I Phase II	Solid tumor	Small cell lung cancer ORR = 0% Squamous cell carcinoma of the head and neck ORR = 0% Gastrointestinal cancers ORR = 0% Gynecologic cancers ORR = 9.1%
**Proteasome Inhibitors**						
**Drug**	**Target**	**ClinicalTrials.gov Identifier**	**Treatment**	**Phase**	**Cancer Type**	**Results ***
Marizomib	Proteasome (20S CS)	NCT00461045	Marizomib	Phase II	Relapsed or relapsed/refractory multiple myeloma	CR = 0%, PR = 0%, SD = 26.7%, PD = 60%, NE = 13.3%
		NCT02330562	Marizomib Bevacizumab	Phase I Phase II	Malignant Glioma Glioblastoma	Marizomib (0.55 mg/m^2^) +Bevacizumab: OR = 50% Marizomib (0.7 mg/m^2^) + Bevacizumab: OR = 33.3% Marizomib (0.8 mg/m^2^) + Bevacizumab: OR = 24.1% Marizomib (1 mg/m^2^) + Bevacizumab: OR = 20%
**DUB Inhibitors**						
**Drug**	**Target**	**ClinicalTrials.gov Identifier**	**Treatment**	**Phase**	**Cancer Type**	**Results ***
Disulfiram	USP21 USP2a	NCT02101008	Disulfiram Chelated zinc	Phase II	Refractory disseminated malignant melanoma	No results available on the CR, PR, SD and OR
		NCT03034135	Disulfiram/copper Temozolomide (TMZ)	Phase II	Recurrent glioblastoma	CR = 0%, PR = 0%

* Results are expressed as percentage of participants with the best overall response according to Response Evaluation Criteria in Solid Tumor (RECIST), version 1.1 guideline: CR (complete response): disappearance of all target lesions; CRi: as per CR but with residual thrombocytopenia (platelet count < 100 × 10^9^/L) or residual neutropenia (ANC < 1.0 × 10^9^/L); PR (partial response): at least 30 percent (%) decrease in sum of diameter of target lesions; PD (disease progression): at least 20% increase in sum of diameter of target lesions; SD (stable disease): neither sufficient shrinkage to qualify for PR nor sufficient increase to qualify for PD; OR (overall response): CR + PR; ORR (objective response rate): the percentage of patients with PR or CR; CR + PR + SD (disease control rate); NE (not evaluable); IV: intravenous; SB: subcutaneous.

## Data Availability

Not applicable.

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
