# Peer review of "Targeting the Ubiquitin–Proteasome System and Recent Advances in Cancer Therapy"

_cells, 2023, doi:10.3390/cells13010029_

Round 1

Reviewer 1 Report

Comments and Suggestions for Authors

The manuscript discusses the therapeutic interventions targeting the ubiquitin-proteasome system (UPS). In doing so the authors describe UPS, ubiquitination as a therapeutic target in cancer treatment, developing strategies, and conclusion. The heavy lifting here is done through brief text mining. That is impressive. The manuscript is well written and accessible and didn't find major flaws in the discussions. However, a couple of minor points to consider are: 1) Critical discussion on the challenge of targeting the Ubiquitin-proteasome system in cancer therapy is missing, 2) The authors must discuss Ubv (ubiquitin variant) used in modulating UPS, 3) the Addition of a figure on different therapeutic modalities of targeting Ubiquitin-proteasome system would be valuable. I believe those additional points would expand the discussion, and make it more structured.

Author Response

Rebuttal letter

Reviewer 1

Comments and Suggestions for Authors

The manuscript discusses the therapeutic interventions targeting the ubiquitin-proteasome system (UPS). In doing so the authors describe UPS, ubiquitination as a therapeutic target in cancer treatment, developing strategies, and conclusion. The heavy lifting here is done through brief text mining. That is impressive. The manuscript is well written and accessible and didn't find major flaws in the discussions. However, a couple of minor points to consider are: 1) Critical discussion on the challenge of targeting the Ubiquitin-proteasome system in cancer therapy is missing, 2) The authors must discuss Ubv (ubiquitin variant) used in modulating UPS, 3) the Addition of a figure on different therapeutic modalities of targeting Ubiquitin-proteasome system would be valuable. I believe those additional points would expand the discussion, and make it more structured.

Reply: We thank the reviewer for the critical reading of manuscript, the positive comments and the suggestions provided to improve it. Please find below the answers to the reviewer’s comments, point by point. We have included a revised manuscript containing the changes marked in red throughout the text.

Comment 1) Critical discussion on the challenge of targeting the Ubiquitin-proteasome system in cancer therapy is missing

Reply: We thank the reviewer for his/her comment. Following the reviewer’s suggestion, we have included a critical discussion about the challenges of UPS targeting in cancer treatment in the section 7 titled “Conclusions and perspectives” at lines 626-647, page 17.

Comment 2) The authors must discuss Ubv (ubiquitin variant) used in modulating UPS

Reply: We thank the reviewer for his/her comment. Following the reviewer’s suggestion, we have included new paragraphs concerning the use of ubiquitin variants as a tool to modulate the function of UPS proteins throughout the text as follows:

  • in section 3 titled “The ubiquitination code” at lines 305-311, page 7;
  • in section 6.1 titled “Targeting ubiquitin-proteasome system to modulate tumor suppressor proteins and oncoproteins” at lines 523-538, page 15;
  • in the section 7 titled “Conclusions and perspectives” at lines 660-665, page 18.

Therefore, the following references have been included in the references list accordingly:

  • Andreas Ernst et al. A strategy for modulation of enzymes in the ubiquitin system. Science. 2013 Feb 1;339(6119):590-5. doi: 10.1126/science.1230161.
  • Tang JQ, Marchand MM, Veggiani G. Ubiquitin Engineering for Interrogating the Ubiquitin-Proteasome System and Novel Therapeutic Strategies. Cells. 2023 Aug 21;12(16):2117. doi: 10.3390/cells12162117.
  • Zhang Y, Zhou L, Rouge L, Phillips AH, Lam C, Liu P, Sandoval W, Helgason E, Murray JM, Wertz IE, Corn JE. Conformational stabilization of ubiquitin yields potent and selective inhibitors of USP7. Nat Chem Biol. 2013 Jan;9(1):51-8. doi: 10.1038/nchembio.1134.
  • Zhang W, Sartori MA, Makhnevych T, Federowicz KE, Dong X, Liu L, Nim S, Dong A, Yang J, Li Y, Haddad D, Ernst A, Heerding D, Tong Y, Moffat J, Sidhu SS. Generation and Validation of Intracellular Ubiquitin Variant Inhibitors for USP7 and USP10. J Mol Biol. 2017 Nov 10;429(22):3546-3560. doi: 10.1016/j.jmb.2017.05.025.
  • Pascoe N, Seetharaman A, Teyra J, Manczyk N, Satori MA, Tjandra D, Makhnevych T, Schwerdtfeger C, Brasher BB, Moffat J, Costanzo M, Boone C, Sicheri F, Sidhu SS. Yeast Two-Hybrid Analysis for Ubiquitin Variant Inhibitors of Human Deubiquitinases. J Mol Biol. 2019 Mar 15;431(6):1160-1171. doi: 10.1016/j.jmb.2019.02.007.

Comment 3) the Addition of a figure on different therapeutic modalities of targeting Ubiquitin-proteasome system would be valuable

Reply: We thank the reviewer for his/her comment. Following the reviewer’s suggestion, an additional figure, named Figure 4, showing the different therapeutic modalities of targeting the UPS has been included in the text in Section 6 titled “Developing strategies to target ubiquitination in cancer therapy”, page 14.

We hope the revised manuscript addresses satisfactorily the reviewer’s comments.

In addition, the figure 3 has been redrawn and its structure and organization have been slightly modified according to the Assistant Editor comment. The modified Figure 3 has been included in the text (page 6) and the figure legend and the text revised, accordingly.

Reviewer 2 Report

Comments and Suggestions for Authors

This is an informative review that updates from similar older reviews particularly regarding recent clinical trials. It is well constructed and I have a few minor comments to improve the readability and information.

1. Pg 1 line 42, as written it suggests that all reactions are 'reversible' which isn't true if the ubiquitination signals for proteasomal destruction. Please reword to make clearer.

2. Pg 2 line 54, please include the classification as to what was used to identify the 'most important' processes (or reword).

3. pg 2 line 58, I believe this sentence would be improved by including 'it' after 'where'

4. Pg 2 line 87-90, the information here should be referenced

5. pg 3 line 109, the reference [2] should be placed prior to 'as discussed in'

6. pg 4 line 161, this last part of the sentence is not clear and could do with rewording

7. Pg 9 Table 1. The title is a bit misleading as targeting ubiquitination is not the same as targeting the proteasome although they are definitely linked. It would be better to work this similar to the subheading 6.1. Also since they are all regarding cancer it would make sense to include 'cancer' in the title somewhere.

8. pg 10 line 408 could be improved with some rewording (more similar to table 2 heading)

9. Table 2 heading includes 'results available' but there are several studies identified that had no results on the measures reported in the table, this needs to be made clearer, additionally as previous the proteasome targeting drugs go against the table title.

10. pg 12, line431, it would be good to include % range for the adverse events reported. Line 437 could be improved with rewording regarding the indications

Author Response

Rebuttal letter

Reviewer 2

Comments and Suggestions for Authors

This is an informative review that updates from similar older reviews particularly regarding recent clinical trials. It is well constructed and I have a few minor comments to improve the readability and information.

Reply: We thank the reviewer for the critical reading of manuscript, the positive comments and the suggestions provided to improve it. Please find below the answers to the reviewer’s comments, point by point. We have included a revised manuscript containing the changes marked in red throughout the text.

Comment 1: Pg 1 line 42, as written it suggests that all reactions are 'reversible' which isn't true if the ubiquitination signals for proteasomal destruction. Please reword to make clearer.

Reply: We thank the reviewer for his/her comment. We agree with reviewer’s comment and the text has been modified accordingly. Please check the lines 42-45, pages 1-2.

Comment 2: Pg 2 line 54, please include the classification as to what was used to identify the 'most important' processes (or reword).

Reply: We thank the reviewer for his/her comment. We agree with reviewer’s comment and the text has been modified accordingly. Please check the line 56, page 2.

Comment 3: pg 2 line 58, I believe this sentence would be improved by including 'it' after 'where'

Reply: We thank the reviewer for his/her comment. We agree with reviewer’s comment and the text has been modified accordingly. Please check the line 59 page 2.

Comment 4: Pg 2 line 87-90, the information here should be referenced

Reply: We thank the reviewer for his/her comment. We agree with reviewer’s comment and the following references have been included in the text at lines 89-90, page 2, accordingly:

  • Reference 54: Adams J, Kauffman M. Development of the proteasome inhibitor Velcade (Bortezomib). Cancer Invest. 2004;22(2):304-11. doi: 10.1081/cnv-120030218.
  • Reference 55: Fostier K, De Becker A, Schots R. Carfilzomib: a novel treatment in relapsed and refractory multiple myeloma. Onco Targets Ther. 2012;5:237-44. doi: 10.2147/OTT.S28911.
  • Reference 56: Robak P, Robak T. Bortezomib for the Treatment of Hematologic Malignancies: 15 Years Later. Drugs R D. 2019 Jun;19(2):73-92. doi: 10.1007/s40268-019-0269-9.
  • Reference 57: Gazzaroli G, Angeli A, Giacomini A, Ronca R. Proteasome inhibitors as anticancer agents. Expert Opin Ther Pat. 2023 Oct 17:1-22. doi: 10.1080/13543776.2023.2272648.

Comment 5: pg 3 line 109, the reference [2] should be placed prior to 'as discussed in'

Reply: We thank the reviewer for his/her comment. We agree with reviewer’s comment and the text has been modified accordingly. Please check the line 111, page 3.

Comment 6: pg 4 line 161, this last part of the sentence is not clear and could do with rewordin

Reply: We thank the reviewer for his/her comment. We agree with reviewer’s comment and the text has been modified accordingly. Please check the lines 163-164, page 4.

Comment 7: Pg 9 Table 1. The title is a bit misleading as targeting ubiquitination is not the same as targeting the proteasome although they are definitely linked. It would be better to work this similar to the subheading 6.1. Also since they are all regarding cancer it would make sense to include 'cancer' in the title somewhere.

Reply: We thank the reviewer for his/her comment. We agree with reviewer’s comment and the title of Table I has been modified accordingly. Please check the title of Table I at line 384, page 9.

Comment 8. pg 10 line 408 could be improved with some rewording (more similar to table 2 heading)

Reply: We thank the reviewer for his/her comment. We agree with reviewer’s comment and the text has been modified accordingly. Please check the lines 419-421, page 10.

Comment 9. Table 2 heading includes 'results available' but there are several studies identified that had no results on the measures reported in the table, this needs to be made clearer, additionally as previous the proteasome targeting drugs go against the table title.

Reply: We thank the reviewer for his/her comment. We agree with reviewer’s comment and the Table II heading has been modified accordingly. Please check the Table II heading at lines 431-433, page 10.

As it concern “results available”, it is related to the results posted on the ClinicalTrials.gov website at the link https://clinicaltrials.gov. Since the end points of each clinical trial differ from study to study, it follows that different parameters have been evaluated in each study, and therefore the measures are not always comparable. Therefore, in Table II the words “results available” have been replaced with “posted results” as indicated at the link https://clinicaltrials.gov.

Comment 10: pg 12, line431, it would be good to include % range for the adverse events reported. 

Reply: We thank the reviewer for his/her comment. The adverse events, experienced by cancer patients in the clinical trials reported in Table II, include multiple disorders affecting diverse organs with a large number of clinical symptoms. As a consequence it is not easy to indicate a range of percentage for these side effects. Therefore, while agreeing with the reviewer's suggestion and appreciating its relevance, we have preferred not to include further information related to this point due to the multiplicity of adverse events. Nonetheless, to meet the reviewer’s comment we have included the following sentence “For more information concerning the treatment-related side effects the readers are referred to the link https://clinicaltrials.gov.” Please check the text at line 459-460, page 14.

We hope that the reviewer will agree with this choice and that he/she is satisfied with these reply.

Line 437 could be improved with rewording regarding the indications

Reply: We thank the reviewer for his/her comment. We agree with reviewer’s comment and the text has been modified accordingly. Please check the lines 451-452, page 13.

We hope the revised manuscript addresses satisfactorily the reviewer’s comments.

In addition, the figure 3 has been redrawn and its structure and organization have been slightly modified according to the Assistant Editor comment. The modified Figure 3 has been included in the text (page 6) and the figure legend and the text revised, accordingly.